# The Impact of Lifestyle on Reproductive Health: Microbial Complexity, Hormonal Dysfunction, and Pregnancy Outcomes

**DOI:** 10.3390/ijms26178574

**Published:** 2025-09-03

**Authors:** Eunice Barraza-Ortega, Bruno Gómez-Gil, Teresa García-Gasca, Dennise Lizárraga, Natalia Díaz, Alejandra García-Gasca

**Affiliations:** 1Laboratory of Molecular Biology and Tissue Culture, Centro de Investigación en Alimentación y Desarrollo, Avenida Sabalo Cerritos sn, Mazatlán 82112, Mexico; eunice.barraza@ciad.mx; 2Laboratory of Microbial Genomics, Centro de Investigación en Alimentación y Desarrollo, Avenida Sabalo Cerritos sn, Mazatlán 82112, Mexico; bruno@ciad.mx; 3Laboratory of Cell Biology, Facultad de Ciencias Naturales, Universidad Autónoma de Querétaro, Avenida de las Ciencias sn, Querétaro 76230, Mexico; tggasca@uaq.edu.mx; 4Laboratory of Embryology, Centro Especializado en Fertilidad y Familia ORIGEN, Mazatlán 82126, Mexico; denniselga@gmail.com; 5Functional Nutriology, Fertility, Hormonal Health and Pregnancy, Plaza Saaghi, Real San Agustín 302, Monterrey 66260, Mexico; natalia@nutriadn.com

**Keywords:** hormonal regulation, human microbiome, lifestyle, pregnancy outcomes, integrative interventions

## Abstract

Endocrine dysfunctions refer to alterations in hormone production, release, or regulation that can significantly impact health. In pregnant women or those planning pregnancy, these conditions may manifest as disorders such as polycystic ovary syndrome, hypothyroidism, endometriosis, gestational diabetes mellitus, and other metabolic issues, which could potentially cause infertility or pregnancy complications. Research and clinical experience indicate that hormones play a crucial role in basic physiology and are essential for overall health and well-being. At the same time, lifestyle—defined as daily habits related to nutrition, exercise, sleep, stress management, and other factors—directly influences microbial composition and hormonal regulation. The human microbiome, a diverse community of microorganisms residing within the human body, plays essential roles in supporting overall health. The increasing prevalence of hormonal disorders, especially in urban populations, has heightened interest in how modern lifestyles—characterised by sedentary habits, chronic stress, imbalanced diets, and inadequate sleep—may contribute to the development or aggravation of these conditions, leading to higher infertility rates or pregnancy complications if untreated. This review investigates the interaction between hormonal dysfunction, the human microbiome, and lifestyle factors, with a focus on their effects on pregnant women and those seeking to conceive. Its purpose is to provide a comprehensive overview of the underlying pathophysiological mechanisms and to examine preventative and therapeutic approaches that could alter these patterns.

## 1. Introduction

Hormonal dysfunctions involve alterations in the production, release, or regulation of hormones essential for maintaining homeostasis and normal body functions. In women, these dysfunctions can result in conditions such as polycystic ovary syndrome (PCOS) or gestational diabetes mellitus (GDM). They may also contribute to infertility, involving both endocrine and non-endocrine factors [1,2]. Furthermore, pregnancy complications like gestational hypertension, premature birth, and preeclampsia are associated with hormonal imbalances, including insulin, cortisol, and thyroid hormones [3,4]. These endocrine disturbances not only affect fertility but also increase the risk of spontaneous abortion [5]. Additionally, insufficient hormonal function can impair placental development and influence the epigenetic programming of the foetus, with long-term implications for offspring health [6]. Such hormonal dysfunctions can be aggravated by factors such as chronic stress, an unhealthy diet lacking in vitamin D, vitamin B12, choline, iron, or magnesium, and a deficiency in physical activity, all of which can disrupt the balance of the body’s microbiome, particularly within the gastrointestinal tract and vagina [7,8].

The human microbiome refers to the complex and dynamic community of microorganisms, including bacteria, viruses, fungi, archaea, and their associated genomes, that inhabit various anatomical sites throughout the body. This microbial ecosystem plays a crucial role in essential physiological functions, including nutrient metabolism, immune system regulation, endocrine modulation, and epigenetic regulation [7,8]. For example, microbial metabolites like short-chain fatty acids (SCFAs)—particularly butyrate and acetate—can modulate DNA methylation by influencing DNA methyltransferases (DNMTs). These epigenetic effects may modify gene expression patterns not only in the mother but also in the developing foetus, potentially shaping long-term health outcomes [9]. SCFAs, mainly acetate, propionate, and butyrate, are produced by gut microbiota during the fermentation of non-digestible fibres and proteins. During pregnancy, SCFAs have multiple roles; they serve as key energy sources, regulate lipid and glucose metabolism, and act as signalling molecules via G-protein-coupled receptors (GPCRs) and histone deacetylases. In GDM, altered SCFA levels are associated with changes in insulin sensitivity, inflammatory responses, and foetal growth outcomes. Butyrate supports gut barrier integrity and promotes fatty acid oxidation, while propionate assists hepatic gluconeogenesis and energy expenditure. Additionally, SCFAs influence immune function and inflammatory regulation in gestational tissues. Notably, their signalling effects can occur through both GPCR-dependent and- independent pathways, including epigenetic mechanisms such as the modulation of histone acetylation and methylation. These mechanisms collectively emphasise the role of SCFAs as mediators of maternal-foetal communication and potential targets for metabolic intervention [10,11].

Recent studies have shown that lifestyle factors, including diet, sleep, stress, physical activity, and environmental exposures, can significantly influence the composition of the microbiome, which, in turn, affects hormonal balance and pregnancy outcomes [12,13]. The association between lifestyle-induced microbiome alterations and reproductive health is an expanding area of research, with evidence suggesting that disruptions in microbial diversity and function may lead to hormonal disorders and adverse pregnancy-related outcomes [14,15]. In addition to biological and behavioural factors, psychological states (e.g., perceived stress, anxiety, depressive symptoms) are associated with adverse obstetric outcomes, likely through stress-related neuroendocrine and immune pathways [16]. We therefore incorporated these mental health factors into our integrative framework.

In 2018, Rothschild et al. [17] stated that the microbiome plays a crucial role in reproductive health, influencing immune, endocrine, and metabolic functions essential for a healthy pregnancy. A balanced microbiome is critical for the development and maturation of the immune system, which helps prevent infections and complications during pregnancy. Research indicates that disruptions in microbial diversity and function may contribute to hormonal imbalances and adverse pregnancy outcomes [18]. Furthermore, the composition of the gut microbiome, especially the bacteriome, has been linked to the regulation of sex hormones such as oestrogen and progesterone, which are essential for ovulation, embryo implantation, and pregnancy [18]. Dysbiosis or changes in the microbiome composition can interfere with these hormonal processes and increase the risk of pregnancy complications, including miscarriage and preeclampsia [19,20].

This article examines the relationship between lifestyle, the microbiome (particularly in the gut and lower reproductive tract), and hormonal dysfunctions that impact reproductive health and pregnancy outcomes. We review recent research demonstrating how alterations in the microbiome can affect the development of hormonal and obstetric complications, and how lifestyle changes can help reestablish microbial equilibrium and enhance reproductive health. Moreover, we discuss how managing factors such as diet, exercise, and stress can positively influence microbial composition and hormonal regulation, thereby reducing the risk of pregnancy complications.

Thus, this review aims to synthesise the mechanistic links between endocrine dysfunction and human microbiome dynamics within the context of lifestyle exposures (nutrition, physical activity, sleep, stress), and to explore preventive and therapeutic strategies relevant to women who are pregnant or seeking to conceive. Specifically, we (i) summarise the pathophysiological pathways connecting hormonal regulation and microbial ecosystems across different body sites, (ii) assess how lifestyle factors influence these interactions and reproductive outcomes, and (iii) highlight clinical implications, evidence gaps, and priorities for future research and clinical translation.

## 2. The Human Microbiome

The human microbiome refers to the collection of microorganisms inhabiting various surfaces and organs of the human body. It encompasses a complex ecosystem, including microbes (bacteria, viruses, archaea, fungi), their genomes, and the environment in which they live. These microbial communities play essential roles in maintaining homeostasis and are actively involved in metabolic, immunological, and endocrine functions [21]. In women, the primary microbial communities are located in the gut, skin, oral cavity, and lower reproductive tract, especially the vagina. However, microbes in breast milk are also significant for newborn health, particularly after caesarean-section deliveries [22]. The gut microbiome is the most diverse and abundant, with its composition linked to numerous systemic functions, including nutrient absorption and regulation of the hypothalamic-pituitary-gonadal (HPG) axis [23].

The HPG axis starts with pulsatile hypothalamic gonadotropin-releasing hormone (GnRH), which stimulates the pituitary to secrete luteinising hormone (LH) and follicle-stimulating hormone (FSH) [24]. GnRH pulse frequency and amplitude regulate the production and release of gonadotropins. LH acts on theca cells to promote androgen biosynthesis, and in the late follicular phase, the pre-ovulatory LH surge triggers ovulation, while FSH encourages granulosa cell proliferation and aromatase-mediated conversion of androgens to oestradiol; after ovulation, luteal progesterone maintains endometrial receptivity and supports early pregnancy [25,26]. Disruptions to GnRH pulsatility or to downstream gonadotropin signalling can impair oocyte maturation, luteal progesterone production, and endometrial receptivity [27].

The vaginal microbiome, primarily composed of *Lactobacillus* species, is crucial for maintaining an acidic environment that prevents infections and promotes fertility [28]. Although the oral bacteriome has traditionally been examined in relation to oral health, recent research indicates that its dysbiosis may contribute to systemic diseases, including preeclampsia [29], and may increase the risk of preterm birth [30]. The interaction between the microbiome and the immune and endocrine systems is mutual and dynamic. The gut bacteriome, for instance, governs the maturation and tolerance of the immune system while also affecting hormone synthesis, such as oestrogen and progesterone, through the regulation of enzymes like β-glucuronidase [8]. Additionally, certain gut bacteria influence metabolite production, such as SCFAs, which are involved in immunoendocrine signalling [31]. These complex interactions clarify how dysbiosis (i.e., an imbalance in the microbiome’s composition and diversity, particularly within the bacteriome) can trigger chronic low-grade inflammation, insulin resistance (IR), thyroid disturbances, and disruptions in the reproductive-hormonal axis, impacting ovarian morphology and the gonadotropin LH/FSH ratio, thereby negatively affecting fertility and pregnancy progression [32].

### 2.1. Diversity of the Human Microbiome

The bacteriome is the most studied component of the human microbiome. Intestinal bacteria play essential roles in fermenting indigestible polysaccharides, producing SCFAs, and regulating immune responses as well as the gut–brain–endocrine axis. In dysbiosis, an altered Firmicutes-to-Bacteroidetes ratio, along with the loss of beneficial bacteria such as *Akkermansia muciniphila* (*A. muciniphila*) and *Faecalibacterium prausnitzii* (*F. prausnitzii*), has been associated with the development of IR, hypertension, and chronic low-grade inflammation. These microbial alterations disrupt metabolic homeostasis and contribute to the pathophysiology of IR-related conditions, including PCOS and type 2 diabetes (T2D) [33,34].

The archaeome, primarily composed of methanogenic archaea, constitutes a small but functionally important component of the human gut microbiome. The most common species, such as *Methanobrevibacter smithii* (*M*. *smithii*) and *Methanosphaera stadtmanae* (*M. stadtmanae*), are involved in removing hydrogen during bacterial fermentation by converting it into methane, which enhances fermentation efficiency [35,36]. This process may improve nutrient absorption and energy use in the host and has been suggested to play a role in the development of obesity and IR under certain conditions [37]. Additionally, archaea can influence immune responses. For example, *M. stadtmanae* strongly activates dendritic cells and triggers the production of pro-inflammatory cytokines in vitro, implying a possible role in intestinal inflammation [38]. However, the diversity within the archaeome is limited compared to other microbial groups, and research has been hampered by the difficulties in cultivating and detecting archaea with standard techniques.

The human virome includes both eukaryotic viruses and bacteriophages. By infecting bacterial hosts, bacteriophages can impact the composition and function of the gut bacteriome, thereby affecting metabolic processes and immune responses. An imbalance in the gut virome has been linked to increased intestinal inflammation and disrupted metabolic homeostasis, which may contribute to diseases such as obesity and IR [39,40,41]. Recent metagenomic studies have shown that the female reproductive tract virome contains both eukaryotic and prokaryotic viruses, which interact dynamically with bacterial communities and the host immune system [42]. For example, bacteriophages (e.g., Siphoviridae and Myoviridae families) can influence microbial composition by shaping bacterial diversity, particularly in conditions such as bacterial vaginosis (BV), where phage-bacteria interactions impact genital inflammation [43]. Likewise, eukaryotic viruses such as human papillomavirus (HPV) and anelloviruses are often found in the lower genital tract and have been linked to inflammation, immune modulation, cancer, and potentially reproductive outcomes like infertility and preterm birth [44,45,46]. Notably, the presence of multiple viral species, rather than a single virus, has been associated with adverse reproductive conditions, emphasising the potential synergistic or dysbiotic effects of the virome on gynaecological health [42].

The mycobiome, although less abundant than the bacteriome, plays a significant role in gut and vaginal health. It mainly comprises yeasts, such as *Candida* spp., and moulds like *Malassezia* spp. and *Debaryomyces* spp. Their overgrowth can compromise the integrity of the intestinal barrier, leading to increased permeability and triggering inflammatory responses. Recent studies indicate that fungal dysbiosis in PCOS patients is characterised by reduced fungal diversity and increased *Candida* spp. abundance, which may disrupt the gut–endocrine axis [47,48]. Changes in the mycobiome have also been linked to hormonal and metabolic profiles, suggesting that fungal communities might contribute to PCOS pathophysiology through effects on cytokine secretion and endocrine regulation [49]. Mechanistically, an overabundance of *Candida albicans* (*C. albicans*) may elevate endotoxin levels and activate pro-inflammatory cytokines, such as interleukin-6 (IL-6) or tumour necrosis factor-alpha (TNF-α), which are known to impair insulin receptor function and glucose metabolism. Furthermore, fungal metabolites can interfere with the gut–endocrine axis by affecting hormonal regulation, potentially intensifying the hormonal imbalances characteristic of PCOS. Therefore, fungal dysbiosis, particularly *C. albicans* overgrowth, acts as a key factor linking gut microbial imbalance with IR and endocrine dysfunction in PCOS patients [5,49,50]. A comprehensive understanding of the microbiome—beyond the bacteriome—is essential for understanding how these microbial communities interact with one another and with the host, thereby influencing metabolic function and endocrine regulation.

### 2.2. The Microbiome and Immuno-Endocrine Interactions

The human microbiome plays a crucial role in regulating the immune and endocrine systems, forming a functional axis called the microbiome–immune–endocrine (MIE) axis. This dynamic interaction regulates inflammatory and metabolic processes, directly influencing hormone production and reproductive health [5]. The MIE axis represents a bidirectional communication system between the body’s resident microorganisms—especially those in the gut and reproductive tract—the immune system, and the endocrine system. This axis forms an integrated functional unit that controls metabolic, immunological, and hormonal processes essential for physiological balance, particularly during reproduction [51].

Metabolic products of the gut microbiome, such as SCFAs, exert immunoregulatory effects by promoting the expansion of regulatory T cells and lowering systemic inflammation through the modulation of pro-inflammatory cytokines, which are involved in hormonal disorders like PCOS and endometriosis [52]. These metabolites also disrupt hormone secretion, including insulin and oestrogen, illustrating the microbiome’s capacity to influence the HPG axis [53].

The HPG axis, which regulates sex hormones such as GnRH, LH, FSH, oestrogen, progesterone, and androgens, is sensitive to immunological and microbial signals. It has been demonstrated that the gut bacteriome can directly influence oestrogen synthesis by modulating the activity of the enzyme β-glucuronidase, which is involved in oestrogen recycling in the liver and intestine [54]. This functional set is known as the oestrobolome. A dysbiotic bacteriome can alter the function of this enzyme, leading to increased or decreased levels of circulating oestrogen, which affects ovulation, embryo implantation, and pregnancy maintenance [53].

Research indicates that dysbiosis, whether in the intestinal or vaginal environments, can lead to endocrine disorders such as PCOS, IR, and thyroid issues, which may worsen chronic inflammatory processes affecting fertility [55,56]. In women with PCOS, for example, a decrease in *Lactobacillus* spp. was accompanied by an increase in pro-inflammatory bacteria, correlating with hyperandrogenism and anovulation [57]. The vaginal bacteriome, usually dominated by *L. crispatus* and *L. jensenii* in healthy conditions, plays a protective role by maintaining an acidic pH and preventing infections that could trigger harmful immune responses during embryo implantation or pregnancy [58]. Changes in the vaginal microbiome may contribute to complications such as premature birth, spontaneous abortion, or preeclampsia [59]. These complex interactions emphasise the importance of microbial balance as a systemic regulator and its dysregulation as a critical factor in hormonal and reproductive disorders (Figure 1).

## 3. Hormonal Regulation and Dysfunctions Related to Reproductive Health

Hormonal regulation is essential for maintaining reproductive health in both women and men. PCOS is one of the most common endocrine disorders in women of reproductive age, characterised by hyperandrogenism, anovulation, and polycystic ovary morphology. This syndrome is closely associated with IR and a low-grade inflammatory state, in which the gut microbiome plays a mediating role [60]. Dysbiosis can increase intestinal permeability, allowing endotoxins to enter the systemic circulation, thereby worsening IR and disrupting hormonal balance [61].

Hypothyroidism also has significant effects on fertility, as the thyroid influences overall metabolism and ovarian follicular development. Low levels of thyroid hormones can interfere with the release of gonadotropins (LH and FSH) from the pituitary gland, disrupt menstrual cycles, decrease libido, and hinder conception [62]. A common outcome of subclinical or clinical hypothyroidism is anovulation, which is linked to disruption of the HPG axis. Elevated thyroid-stimulating hormone (TSH) levels can also raise prolactin levels, which inhibit the release of GnRH and prevent ovulation [63]. Moreover, women with hypothyroidism may experience luteal phase abnormalities characterised by low progesterone levels [64]. Infertility related to hypothyroidism may also be caused by endometrial dysfunction, which affects uterine receptivity and early embryonic development [65]. Additionally, low progesterone levels impair endometrial receptivity and early embryonic development, increasing the risk of spontaneous miscarriage [66,67]. 

Untreated hypothyroidism during pregnancy has been associated with adverse outcomes such as premature birth, intrauterine growth restriction, and impaired foetal neurocognitive development, highlighting the importance of early diagnosis and appropriate treatment, especially for women trying to conceive [68]. Levothyroxine supplementation in women with hypothyroidism has been shown to restore ovulation and improve pregnancy rates, even in assisted reproduction cycles. Therefore, monitoring the complete thyroid profile—including TSH, free T4, free T3, and thyroid autoantibodies—is vital in managing female infertility related to thyroid dysfunction [69,70]. Current clinical guidelines recommend maintaining serum TSH levels below 2.5 mIU/L during pregnancy to reduce the risk of miscarriage and other gestational complications. Maintaining TSH within this range is crucial for optimal maternal and foetal outcomes, emphasising the need for early thyroid function assessment and appropriate adjustment of levothyroxine doses in women planning pregnancy or undergoing fertility treatments [69,71].

Insulin resistance is a metabolic condition in which peripheral tissues, such as muscle, liver, and adipose tissue, exhibit a reduced response to insulin’s effects, leading to pancreatic compensation through increased insulin secretion (hyperinsulinemia). This alteration is not only associated with metabolic diseases like T2D and obesity but also significantly affects female reproductive function [72]. One of the most common outcomes of IR in women of reproductive age is linked to PCOS. In this context, hyperinsulinemia contributes to ovarian dysfunction via several pathways. First, it directly stimulates ovarian theca cells to produce more androgens, resulting in hyperandrogenism. Second, it decreases hepatic production of sex hormone-binding globulin (SHBG), which raises circulating free testosterone levels [73]. This condition causes chronic anovulation, menstrual irregularities, and infertility [74]. Moreover, high insulin levels disrupt signalling in the HPG axis, leading to an imbalance in the cyclical release of LH and FSH, both of which are essential for follicular maturation and ovulation. This can cause a significant reduction in fertility and may require specialised treatments to achieve pregnancy [75].

In pregnancy, IR is a significant risk factor for obstetric complications such as gestational diabetes, preeclampsia, premature birth, and intrauterine growth restriction. Additionally, women with hyperinsulinemia face a higher risk of spontaneous miscarriage, likely due to changes in endometrial receptivity, chronic low-grade inflammation, and vascular dysfunction [76]. Therefore, assessing insulin sensitivity should be a key part of the workup for female infertility, especially in women with PCOS or signs of hyperandrogenism. Treatment options may include lifestyle changes and the use of insulin sensitisers, such as metformin, which have shown both metabolic and reproductive benefits [77]. Moreover, myo-inositol has become recognised as a safe and effective supplement that improves insulin resistance and ovulatory function. It is also used during pregnancy to help prevent GDM, thus reducing maternal and foetal complications [78].

## 4. The Interaction Between Hormones and Microorganisms in Fertility and Pregnancy

Hormonal balance is essential during pregnancy for supporting implantation, placental function, and foetal development. Progesterone maintains the endometrium, while oestrogen regulates uterine blood flow [67,79]. Hormonal imbalances during pregnancy can result in conditions such as preeclampsia, premature birth, intrauterine growth restriction, and spontaneous miscarriage [32,80].

The microbiome is increasingly recognised for its role in hormonal regulation. A key example is the oestrobolome, a part of the gut bacteriome that influences oestrogen metabolism. These bacteria produce enzymes such as β-glucuronidase, which break down oestrogen conjugates and enable their systemic recirculation. A dysfunctional oestrobolome can alter circulating oestrogen levels, impacting ovulation and endometrial development [54]. Moreover, the gut bacteriome affects cortisol production through the gut–brain axis. An imbalance in this system can lead to the overactivation of the hypothalamic–pituitary–adrenal (HPA) axis, disrupting gonadal function and reducing fertility [81]. For clarity, the HPA axis involves the hypothalamus releasing corticotropin-releasing hormone (CRH), which prompts the pituitary to secrete adrenocorticotropic hormone (ACTH), thereby stimulating the adrenal glands to produce cortisol [82]. When chronically activated, this stress response can interfere with neuroendocrine signalling that supports ovulation, luteal function, and early pregnancy maintenance [83,84].

Recent research suggests that specific gut bacteria can impact insulin sensitivity and leptin signalling, thereby linking metabolic processes to hormonal regulation [85]. Psychological stress activates the HPA axis and raises maternal and placental CRH signalling, with downstream effects on cortisol levels [86,87]. Observational and meta-analytical studies associate antenatal stress and depression with increased risks of preterm birth and low birthweight [88]. Emerging evidence proposes a ‘gut–germline axis’ whereby microbiome-driven signals influence germ cell programming and, ultimately, offspring health. Preclinical data suggest that paternal microbiome perturbations can alter the germline molecular cargo, impacting embryo and offspring outcomes. Although direct human evidence related to pregnancy remains limited, these findings support considering paternal microbiome exposures as a potential upstream factor in reproductive outlook and perinatal health [89].

### The Microbiome-Immune-Endocrine Axis

The MIE axis denotes a complex bidirectional communication network between the human body’s microorganisms, the immune system, and the endocrine system. This interaction is crucial for maintaining hormonal balance and reproductive health, especially during pregnancy (Table 1 and text below).

Metabolic–endocrine regulators (insulin, stress, inflammation). Members of the gut bacteriome modulate metabolic and stress axes with downstream reproductive effects [90,91,95]. *A. muciniphila* improves insulin sensitivity and reduces systemic inflammation—changes particularly relevant in PCOS-related anovulation [90]. *F. prausnitzii* produces butyrate and increases IL-10, reducing inflammatory tone and stress responses linked to cortisol [91]. Additionally, *Bifidobacterium longum* (*B. longum*) has been associated with lower cortisol levels and improved ovulatory outcomes in fertility contexts [95].

Oestrogen and androgen metabolism (oestrobolome, hepatic clearance). Microbial β-glucuronidase activity (e.g., *Clostridium* spp.) promotes enterohepatic recirculation of oestrogens, potentially increasing exposure to oestrogen-dependent conditions [92]. *Bacteroides* spp. influence hepatic steroid metabolism and may alter androgen availability, with implications for hyperandrogenism in PCOS [96]. These pathways collectively link gut dysbiosis with endocrine imbalance [96].

Upper-tract pathogens and fertility. *Chlamydia trachomatis* (*C. trachomatis*), *Mycoplasma genitalium* (*M. genitalium*), and *Ureaplasma urealyticum* (*U. urealyticum*) cause chronic inflammation and tissue remodelling in the endometrium and fallopian tubes, leading to tubal scarring, infertility, ectopic pregnancy, and preterm complications; *Toxoplasma gondii* (*T. gondii*) disturbs placental Th1/Th2 balance with risks for miscarriage and foetal compromise [105,106,107,108,109,110,111].

Vaginal ecosystem stability and implantation. A *Lactobacillus* spp.-dominant community maintains a low pH and mucosal tolerance, supporting implantation and reducing the risk of infection and preterm birth [58]. Species such as *L. rhamnosus* and *L. reuteri* strengthen urogenital defences—limiting Group B *Streptococcus* and modulating cytokines—thereby supporting endometrial receptivity during pregnancy [102,103,104,112]. The hormonal environment (notably oestrogens) influences *Lactobacillus* spp. abundance and function throughout the menstrual cycle and pregnancy [113,114].

Pathobionts and obstetric risk. The over-representation of *Prevotella* spp. and bacterial vaginosis-associated taxa, such as *Gardnerella vaginalis* (*G*. *vaginalis*), disrupts cervical mucus and local mucosal immunity, linking to dysbiosis, subfertility, and an increased risk of preterm birth [57,93,115,116]. *Streptococcus agalactiae* (*S. agalactiae*) colonisation, if uncontrolled, increases the risk of perinatal infection and chorioamnionitis [92].

Mycobiome and virome contributions. *C. albicans* and *Malassezia* spp. can amplify mucosal inflammation, undermining epithelial integrity and endocrine–metabolic balance [48,97,98]. The virome also influences bacterial communities; for example, human papillomavirus (HPV) disrupts mucosal defences and has been associated with infertility and adverse implantation [99,117]. Meanwhile, anelloviruses and bacteriophages are associated with *Lactobacillus*-depleted states and an increased risk of BV, with potential implications for pregnancy outcomes [43,99,100]. This is consistent with broader evidence on virus–bacteriome–immunity interactions [101].

## 5. Influence of Lifestyle on Gut Microbiome and the Endocrine System

The modern lifestyle has a significant impact on both the human microbiome and endocrine function. Elements such as diet, stress, physical activity, sleep, and the use of antibiotics or hormonal medications directly influence these systems, impacting reproductive health and pregnancy outcomes.

### 5.1. Diet and Exercise

Diet is a key factor influencing the diversity and function of the intestinal microbiome. High-fibre diets promote the growth of beneficial bacteria, such as *F. prausnitzii* and *A. muciniphila*, which are involved in producing SCFAs, including butyrate, that possess anti-inflammatory properties and help regulate the HPA axis [118]. In contrast, a diet rich in saturated fats and simple sugars promotes the growth of pro-inflammatory bacteria, such as *Bilophila wadsworthia* (*B. wadsworthia*), which can damage the intestinal barrier, increase oxidative stress, and impair hormonal signalling [119]. 

The Mediterranean diet (MedDiet)—characterised by high consumption of fruits, vegetables, whole grains, legumes, nuts, and olive oil, moderate intake of fish and poultry, and low consumption of red meat and sweets—has consistently been linked to improved ovulatory function and a lower risk of anovulatory infertility. The Nurses’ Health Study II [120], found that women with the highest adherence to a “fertility diet” pattern—rich in monounsaturated fats, vegetable protein, low-GI carbohydrates, high-fat dairy, multivitamins, and plant-based iron—had a 66% lower risk of ovulatory infertility compared to those with the lowest adherence (RR 0.34; 95% CI 0.23–0.48). Similarly, increased intake of vegetable protein and reduced dietary glycaemic load were each linked to a lower risk of ovulatory infertility (RR 0.56 per 10 g/day of cereal fibre; RR 1.91 for the highest versus lowest total carbohydrate intake) [121]. Recent meta-analyses and cohort reviews have confirmed these findings, indicating that adherence to MedDiet principles (emphasising unsaturated fats, fibre, and antioxidants) can boost ovulation, enhance oocyte quality, support embryo development, and increase clinical pregnancy and live birth rates, particularly in assisted reproduction technologies (ARTs) [122]. Notably, reducing the consumption of red and processed meats has been recognised as a protective factor for reproductive health. High intake of these foods has been linked to hormonal imbalance, oxidative stress, and ovulatory issues. In the Nurses’ Health Study II, consuming one additional daily serving of red or processed meat—while maintaining total calorie intake—was associated with a 32% increased risk of ovulatory infertility (relative risk [RR] 1.32; 95% CI 1.08–1.62) [120].

In a separate prospective cohort, higher preconception red meat intake among women undergoing ARTs was linked to impaired embryo development and lower clinical pregnancy rates, suggesting that diets high in meat may negatively affect both natural and assisted conception [123]. These findings support including the Mediterranean diet—which emphasises plant-based proteins and limits red meat—as a modifiable factor to improve ovulatory health.

Moderate, regular exercise has been shown to positively influence the gut microbiota, increase bacterial diversity, and improve the Firmicutes/Bacteroidetes ratio [124]. These microbial changes boost the production of SCFAs, strengthen gut barrier integrity, and reduce inflammation—mechanisms that collectively enhance insulin sensitivity and help sustain a more balanced hormonal profile by regulating glucoregulatory and reproductive hormones [125,126]. Furthermore, it has been demonstrated that exercise lowers chronic cortisol levels and improves HPA axis function, thereby positively impacting ovulation and the production of sex hormones such as oestrogen and progesterone [82,127,128]. Recent studies have shown that strength training can elevate LH levels and increase sensitivity to FSH, thereby supporting ovulation and stabilising the menstrual cycle [129]. In women with PCOS, strength training has been observed to decrease circulating androgen levels, such as free testosterone, and improve the LH/FSH ratio, thus fostering a more favourable endocrine environment for fertility [130]. While resistance training may acutely increase anabolic hormones, such as growth hormone (GH) and testosterone, especially following exercise sessions, its long-term impact in women with hyperandrogenism appears to be regulatory, helping to lower excessive androgen levels and cortisol, ultimately supporting ovulation and implantation [131,132].

Strength training activates molecular pathways, such as the phosphatidylinositol 3-kinase/protein kinase B (PI3K/AKT) and AMP-activated protein kinase (AMPK), in muscle tissue, which significantly enhances insulin-independent glucose uptake. This is essential in women with IR, as hyperinsulinemia negatively affects ovarian androgen production and promotes anovulation [133]. Moreover, muscle mass serves as a metabolic reservoir, supporting a more stable long-term hormonal profile. Although intense exercise temporarily raises cortisol levels, progressive and well-periodised strength training helps modulate the HPA axis response, lowering basal stress and enhancing hormonal resilience. This cortisol regulation decreases systemic inflammation and restores HPG axis function, aiding processes such as follicular maturation and embryo implantation [134]. 

The combination of a sufficient diet rich in fibre and strength training has been associated with increased intestinal microbial diversity, as it promotes the growth of species such as *F. prausnitzii* and *Roseburia* spp., which produce butyrate. These bacteria have anti-inflammatory properties and stimulate the secretion of glucagon-like peptide-1 (GLP-1) and other intestinal hormones that interact with the endocrine axis [135]. This is relevant because the bacteriome directly influences the bioavailability of oestrogens and systemic hormonal balance (Figure 2).

Diet and exercise remain essential for promoting diversity in the intestinal microbiome and supporting endocrine health. Combining strength training with a high-fibre diet boosts the presence of butyrate-producing bacteria such as *F. prausnitzii* and *Roseburia* spp., which stimulate glucagon-like peptide-1 (GLP-1) secretion and help reduce inflammation [136]. Meanwhile, the ‘30 plant foods per week’ concept—developed by Tim Spector—emphasises the importance of dietary diversity through a wide intake of fruits, vegetables, legumes, whole grains, seeds, nuts, herbs, and spices. This approach notably improves microbial richness and resilience [137]. Certain prebiotic fibres, including inulin, resistant starch, and arabinoxylans—naturally found in oats, legumes, onions, and chicory root—have been demonstrated to selectively support *F. prausnitzii*, *A. muciniphila*, and *Roseburia* spp., thereby fostering gut barrier function and insulin sensitivity [138,139].

Additionally, supplementing with probiotic strains such as *L. acidophilus*, *L. casei*, and *B. bifidum* has been shown to benefit women with GDM, notably improving fasting blood glucose, HOMA-IR, insulin, triglycerides, and VLDL levels after six weeks of use [140]. Fermented foods, such as yoghurt, kefir, kimchi, and miso, also provide live microbes and postbiotic metabolites that support immune regulation and microbial diversity during pregnancy. Overall, a strategy combining plant-diverse, fibre-rich nutrition, targeted probiotics, and fermented foods may positively influence the gut–endocrine axis, enhance ovulation, and help prevent metabolic conditions like GDM [141].

### 5.2. Chronic Stress

The gut–brain axis is a bidirectional communication system that connects the central nervous system, the enteric nervous system, and the gut microbiome through hormonal, immune, and nerve pathways. Chronic stress can greatly disrupt this axis, impacting gut homeostasis, epithelial barrier permeability, microbial composition, and hormonal regulation related to reproductive health [142]. 

During stressful situations, the HPA axis is activated, leading to increased glucocorticoid release, including cortisol. This prolonged elevation of cortisol has immunomodulatory effects, modifying cytokine production and decreasing gut microbial diversity, particularly that of beneficial species such as *Lactobacillus* spp. and *Bifidobacterium* spp. It also promotes the overgrowth of pathogenic bacteria such as *Clostridium* spp. and *Escherichia coli* (*E. coli*) [81,143]. These changes affect the production of neurotransmitters like serotonin, gamma-aminobutyric acid (GABA), and dopamine, which are partly synthesised in the gut and influence stress responses and mood [144,145]. 

Systemic inflammation caused by dysbiosis can alter cortisol sensitivity, negatively impacting the HPA axis [146]. Additionally, stress-induced dysfunction of the gut–brain axis has been associated with hormonal imbalances that affect female reproductive health. Chronic stress has been shown to contribute to hypothalamic functional anovulation, decrease oestradiol and progesterone levels, and inhibit the pulsatile release of GnRH, LH, and FSH, thereby interfering with ovulation and fertility [147]. Moreover, elevated cortisol levels from chronic HPA axis activation can suppress TSH secretion and reduce the peripheral conversion of thyroxine to triiodothyronine, potentially leading to subclinical hypothyroidism and further disrupting reproductive hormonal balance [148]. These responses are exacerbated by intestinal inflammation and insulin resistance linked to dysbiosis [149].

### 5.3. Environmental Toxins

Exposure to environmental toxicants such as pesticides and endocrine-disrupting chemicals (EDCs) significantly contributes to oxidative stress and reproductive dysfunction [150]. Pesticides increase reactive oxygen species (ROS) and deplete antioxidants such as glutathione (GSH) and vitamin E, while polychlorinated biphenyls (PCBs) cause oxidative stress through endothelial damage and membrane disintegration. EDCs like phthalates and bisphenol A (BPA) are linked to hormonal disruption, ROS production, and sperm apoptosis, and have also been associated with recurrent miscarriage and unexplained infertility in women.

Occupational exposure, particularly in agriculture and industries related to polyvinyl chloride (PVC), poses an elevated reproductive risk [151,152]. Furthermore, genetic polymorphisms in detoxification enzymes—such as glutathione-S-transferase *GSTM1* and *GSTT1* (Phase II) and cytochrome P-450 *CYP17A1* (Phase I)—may reduce the efficiency of xenobiotic metabolism and antioxidant defence, thereby increasing individual susceptibility to oxidative stress-related reproductive damage [6,153]. In addition, converging human and experimental evidence indicates that both pesticides and EDCs perturb the gut microbiota, reducing diversity and shifting community composition (e.g., *Akkermansia* spp., *Bacteroides* spp.), with downstream effects on host metabolism, inflammatory tone, and systemic immunity [154,155,156]. These dysbiotic changes can also interfere with endocrine signalling via the endobolome (the microbiome-encoded pathways that metabolise steroid hormones and xenobiotics) [157], and EDC exposure has been linked to hypothalamic inflammation and adverse reproductive outcomes [158]. Together, these microbiota-mediated mechanisms provide an additional route by which toxicant exposure can exacerbate oxidative stress and contribute to infertility, pregnancy complications, and broader endocrine dysfunction [154,155,157,158].

### 5.4. Drugs and Antibiotics

Chronic use of antibiotics, oral contraceptives, nonsteroidal anti-inflammatory drugs (NSAIDs), and proton pump inhibitors significantly modifies the composition and diversity of the gut bacteriome, often causing dysbiosis [159]. Antibiotics decrease microbial diversity, leading to a reduction in the production of SCFAs, which are essential for controlling inflammation and peripheral hormonal signalling. Similarly, NSAIDs have been shown to disrupt gut microbial balance, affecting bacterial diversity and metabolic activity [160,161,162]. Moreover, oral contraceptives can induce changes in the oestrobolome, influencing the enterohepatic recirculation of these hormones and increasing the risk of hormonal imbalances [163].

### 5.5. Sleep and Circadian Rhythms

Sleep deprivation disrupts circadian rhythms, which directly affects both the endocrine system and the gut microbiome. Recent studies have shown that sleep disruption reduces the abundance of key bacteria, such as *Roseburia* spp. and *Lactobacillus* spp., while increasing cortisol levels and decreasing melatonin production, a hormone involved in ovarian regulation and the synchronisation of the reproductive hormonal axis [164]. Additionally, chronic sleep disorders interfere with metabolic hormones, raising ghrelin and lowering leptin, which triggers hunger and increases caloric intake, and also impairs insulin sensitivity, contributing to glucose intolerance and metabolic dysfunction [165,166,167].

### 5.6. Mental States (Stress, Anxiety, Depressive Symptoms)

Psychological distress during pregnancy—including stress, anxiety, or depressive symptoms—can disrupt endocrine–immune interactions (including HPA axis function) and has been linked in meta-analyses to preterm birth and low birthweight [88]. Mechanistically, increased maternal and placental CRH, together with related inflammatory signalling, may influence foetal development and precipitate labour. Neuroimaging studies further connect higher maternal distress with alterations in foetal brain structure and connectivity [86,168]. Additionally, emerging evidence suggests that maternal psychological stress is associated with gut microbiota perturbations—lower diversity, increased intestinal permeability, and heightened systemic inflammation—which may impact foetal development through maternal–foetal gut–brain pathways [143,169,170]. Overall, these data support the importance of considering psychological health alongside sleep, diet, and physical activity in preconception and antenatal care [171].

## 6. Microbiome Changes During Pregnancy

During pregnancy, various changes occur in a woman’s hormonal, immunological, and metabolic environment, which also significantly influence the composition of the intestinal and vaginal microbiomes. At the intestinal level, a reduction in microbial diversity has been documented, with a relative increase in the phyla Proteobacteria and Actinobacteria. Meanwhile, the abundance of Firmicutes and Bacteroidetes decreases at certain stages [18]. These physiological changes respond to an adaptive need to promote mild intestinal inflammation, enabling energy storage and the immunological tolerance necessary for foetal development [18,172].

Within the vaginal microbiome, a shift towards a community dominated by *Lactobacillus* spp., especially *L. crispatus*, *L. iners*, and *L. jensenii*, is also observed, which helps preserve a protective acidic pH against pathogens [59]. This stable colonisation is vital during pregnancy, as it defends against ascending infections, such as bacterial vaginosis or *Gardnerella vaginalis* infections, which can threaten the normal course of pregnancy [59,173,174]. In addition to bacteria, fungi are also part of the vaginal microbiome, with *Candida* species—particularly *C. albicans*—being the most common. Although generally commensal, dysbiosis or immunological changes during pregnancy can promote fungal overgrowth, leading to vulvovaginal candidiasis, which has been linked to preterm birth and other obstetric complications [175,176].

### 6.1. Dysbiosis in Pregnancy

Significant obstetric complications can occur when microbial adaptations are pathologically disrupted. Intestinal dysbiosis, characterised by a predominance of pro-inflammatory bacteria such as *E. coli* or *Shigella* spp., can trigger a strong immune response and is linked to adverse metabolic conditions, including GDM, preeclampsia, and excessive maternal weight gain [20,177].

On the other hand, vaginal dysbiosis is strongly linked to preterm delivery, premature rupture of membranes, and low birth weight. The presence of microbial communities not dominated by *Lactobacillus* spp., especially those enriched with *Gardnerella* spp., *Atopobium* spp., *Prevotella* spp., and *Mobiluncus* spp., has been recognised as a risk biomarker for these complications [111,178]. Additionally, oral dysbiosis, particularly with increased colonisation by *Fusobacterium nucleatum* (*F. nucleatum*), has been associated with adverse pregnancy outcomes, including preterm birth and stillbirth. This pathogen can translocate from the oral cavity to the placenta, triggering inflammation and foetal injury [179,180].

### 6.2. Role of the Maternal Microbiome in Foetal Development

The maternal microbiome directly impacts foetal immunological, neurological, and metabolic development. While some studies have discredited the existence of an active resident microbiota in the placenta or amniotic fluid [181], new evidence suggests an alternative communication route between the maternal microbiome and the foetus [182]. Bacterial extracellular vesicles (EVs) are small spherical structures released by bacteria, made up of lipids, proteins, and nucleic acid fragments. These vesicles can cross biological barriers, including the placental barrier, serving as signalling vehicles between the maternal microbiome and the developing foetal immune system [183]. Unlike active colonisation, this process represents a form of non-infectious molecular communication that potentially influences foetal immune development and the neuroendocrine axis.

Recent research has identified bacterial EVs in maternal blood and placental tissues, suggesting that these particles may transport metabolites and immunologically relevant information that influence the maturation of the foetal immune system [184]. These interactions might prepare the foetus to face the postnatal microbial environment, promoting suitable immunological tolerance and modulating the expression of genes related to the HPA axis and stress response. This finding is crucial, as it opens new pathways for studying how the maternal microbial environment, even without live bacteria in the uterus, can actively influence the immuno-endocrine programming of the foetus, with potential long-term effects on the newborn’s health. Moreover, the mother’s microbiome shapes the initial colonisation of the newborn during vaginal delivery and through breast milk. This early microbiome is essential for the long-term health of the newborn, influencing susceptibility to inflammatory, metabolic, and autoimmune diseases in childhood and adulthood [185].

## 7. Preventive and Therapeutic Approaches Through Lifestyle Modifications

### 7.1. Diet and Exercise-Based Interventions

Diet represents one of the essential pillars for modulating the MIE axis. For instance, the Mediterranean diet benefits reproductive health and hormonal balance due to its high fibre content, healthy fats (such as omega-3 fatty acids), antioxidants, and anti-inflammatory compounds [186]. This dietary pattern promotes gut microbiota diversity, reduces systemic inflammation, and enhances insulin sensitivity, which are essential for conditions such as PCOS and IR [187]. Similarly, the therapeutic use of probiotics (live microorganisms) and prebiotics (substrates that support their growth) has gained popularity in the gynaecological field. *Lactobacillus rhamnosus* (*L. rhamnosus*) and *Bifidobacterium bifidum* (*B. bifidum*) have been shown to benefit vaginal balance and reduce inflammatory processes that impact fertility [188]. Additionally, certain strains have been shown to positively influence the regulation of the HPA axis and modulate cortisol levels [189].

Exercise, particularly strength training, not only improves body composition and insulin sensitivity but also influences the hormonal profile. Recent studies have demonstrated that resistance training increases GH production while lowering cortisol levels, creating a more favourable endocrine environment for ovulation and implantation [131]. Additionally, exercise enhances the gut microbiome by promoting bacterial diversity and increasing the production of metabolites such as SCFAs, which have immunoregulatory and anti-inflammatory effects [190].

### 7.2. Personalised Strategies and Microbiome-Targeted Therapies

Recent advances in omics technologies have enabled a more precise understanding of the human microbiome and its connection with the endocrine system and reproductive health [191]. Building on this scientific basis, personalised therapeutic approaches have developed that aim to directly modify the host’s microbial ecosystem to influence hormonal, immunological, and metabolic processes [192,193,194].

Personalised medicine is emerging as a powerful tool to address hormonal dysfunctions from an integrative perspective [195,196]. Each individual has a unique microbial signature that influences their response to nutrients, medications, and hormonal factors [161,193]. This individuality has driven the development of personalised diets based on faecal microbiota analysis, where specific imbalances are identified. Dietary plans are designed to favour the growth of beneficial bacteria, such as *F. prausnitzii* or *A. muciniphila*, which are associated with insulin sensitivity and effective immune function [197,198]. These interventions enhance microbial composition and induce epigenetic changes that regulate the expression of genes related to inflammation and fertility [199].

Unlike generic probiotic supplementation, personalised approaches aim to use specific strains tailored to an individual’s gut environment and their particular functions. For instance, *L. crispatus* and *L. jensenii* have been extensively studied in women experiencing recurrent vaginal infections or urogenital microbial imbalances, demonstrating improvements in vaginal epithelial integrity and a reduction in pH, which helps create an optimal environment for fertilisation and implantation [93,200]. Faecal microbiota transplantation has traditionally been employed to treat *Clostridioides difficile* (*C. difficile*) infections; however, recent research indicates its potential to address severe dysbiosis associated with infertility, endometriosis, or metabolic syndrome [201].

Integrating artificial intelligence (AI) with microbiome data enables the development of predictive algorithms that identify the risk of hormonal dysfunction (e.g., PCOS), infertility/ART outcomes, and pregnancy complications (e.g., preterm birth, GDM) [202,203,204]. These algorithms incorporate variables such as diet, hormone levels, inflammatory biomarkers, and microbial composition, facilitating the creation of personalised intervention programmes [17]. Recently, AI-based approaches have started to include nutrigenetic/nutrigenomic data (e.g., single-nucleotide polymorphisms (SNPs) or polygenic risk) alongside gut microbiome features to develop multi-omic predictors for diet and health [205]. This combination enables highly personalised dietary advice—including targeted prebiotic interventions—aimed at promoting the growth of beneficial bacterial taxa based on each individual’s genetic and microbial profile [193,206]. Additionally, emerging therapies such as postbiotics (bioactive metabolites produced by microorganisms) are gaining interest due to their capacity to modulate pathways, including interleukins, cortisol, oestrogens, and SCFAs, without requiring direct bacterial colonisation [207]. Together, these personalised strategies represent a progression of conventional medicine towards an integrative and preventive model, where understanding the microbiome and its interaction with the endocrine system facilitates early and precise interventions, enhancing fertility, pregnancy, and long-term hormonal health outcomes [208].

This approach aligns with the P4 medicine model—Predictive, Preventive, Personalised, and Participatory—which highlights the utilisation of systems biology, digital tools, and patient engagement to anticipate and manage disease before clinical symptoms appear, particularly in complex conditions such as hormonal and reproductive disorders [209] (Figure 3).

### 7.3. Male Contributions to Fertility and Pregnancy Outcomes

Although this review focuses on women’s health, reproductive success is also influenced by male factors. Paternal metabolic and endocrine health (such as obesity, insulin resistance, thyroid dysfunction) is linked to poorer semen quality, sperm DNA damage, and epigenetic changes, which can cause subfertility, miscarriage, and adverse perinatal outcomes [210,211,212,213,214]. Both seminal and gut microbiomes may impact local immune responses and gamete function [215]. Observational and sequencing studies demonstrate that seminal dysbiosis is associated with altered semen parameters (such as count, motility, and morphology) and increased sperm DNA fragmentation [216,217,218]; for instance, the presence of *L. iners* is related to reduced motility, whereas enrichment of *Pseudomonas stutzeri* (*P. stutzeri*) and *P. fluorescens* (and lower *P. putida*) associates with low sperm concentration [216]. Emerging clinical and preclinical studies suggest that gut-derived metabolites—SCFAs (e.g., butyrate) and phenolic acids (e.g., 3-hydroxyphenylacetic acid)—and bile acid signalling can influence sex hormone levels, testicular metabolism, and spermatogenesis [219,220].

In animal models, butyrate enhances semen quality and testosterone levels, while 3-HPAA rescues spermatogenesis via the GPX4/ferroptosis pathway [221,222]. Mendelian randomisation analyses further indicate a Ruminococcaceae NK4A214 Group → bile acids/vitamin A axis connection, linking gut microbial profiles to male reproductive outcomes and infertility risk [223,224]. Additionally, preclinical studies support a “gut–germline axis,” where disruptions to the paternal microbiome can reprogramme sperm small-RNA cargo and adversely affect offspring growth and survival [89]. Optimising diet, physical activity, sleep, and stress management before conception—and avoiding smoking, excess alcohol, and anabolic-androgenic steroids—can improve semen quality and fecundability [225,226,227].

### 7.4. Clinical Assessment and Intervention of Mental States

In accordance with major guidelines, routine screening for perinatal depression and anxiety at the initial prenatal appointment, later in pregnancy, and postpartum is advised, using validated tools, with stepped-care pathways for treatment [228]. Non-pharmacological options, such as dietary/lifestyle intervention, cognitive-behavioural therapy, and structured psychosocial support, are effective for many patients; pharmacotherapy may be indicated for moderate to severe illness after a risk–benefit discussion [229].

## 8. Conclusions

Throughout this article, we have examined how lifestyle significantly impacts the human microbiome, endocrine function, and reproduction. Recent scientific evidence shows that diet, exercise, stress management, and daily habits can influence the intestinal, vaginal, and oral microbiomes, with direct effects on hormone production and reproductive health [23,230]. These interactions are governed by complex immunoendocrine pathways, including the production of SCFAs, oestrogen metabolism via the oestrobolome, and HPA axis regulation, all of which are affected by microbial composition and diversity [231,232]. Common hormonal disorders such as PCOS, hypothyroidism, and IR are closely linked to microbiome changes. This dysbiosis can adversely affect fertility, ovulation, endometrial health, and pregnancy outcomes, including a heightened risk of miscarriage, preeclampsia, and preterm birth [72,233].

From a clinical and public health perspective, these findings highlight the urgency of integrating evidence-based, personalised lifestyle strategies into both preventive and therapeutic approaches to reproductive disorders. An adequate diet, along with the use of probiotics, prebiotics, postbiotics, and resistance/strength training, is emerging as a cost-effective and accessible intervention to improve hormonal and reproductive health in women [18,234]. Moreover, the increasing interest in microbiome-targeted therapies, such as modulating hormonal and immunological pathways without the need for direct bacterial colonisation, opens new horizons for reproductive medicine. Incorporating the microbiome, especially the bacteriome, as a diagnostic biomarker and therapeutic target can transform gynaecological, obstetric, and endocrine interventions. Therefore, an integrated approach that considers the microbiome-immune-endocrine axis and lifestyle factors is essential for improving reproductive outcomes and promoting women’s overall health throughout their life cycle.

## Figures and Tables

**Figure 1 ijms-26-08574-f001:**
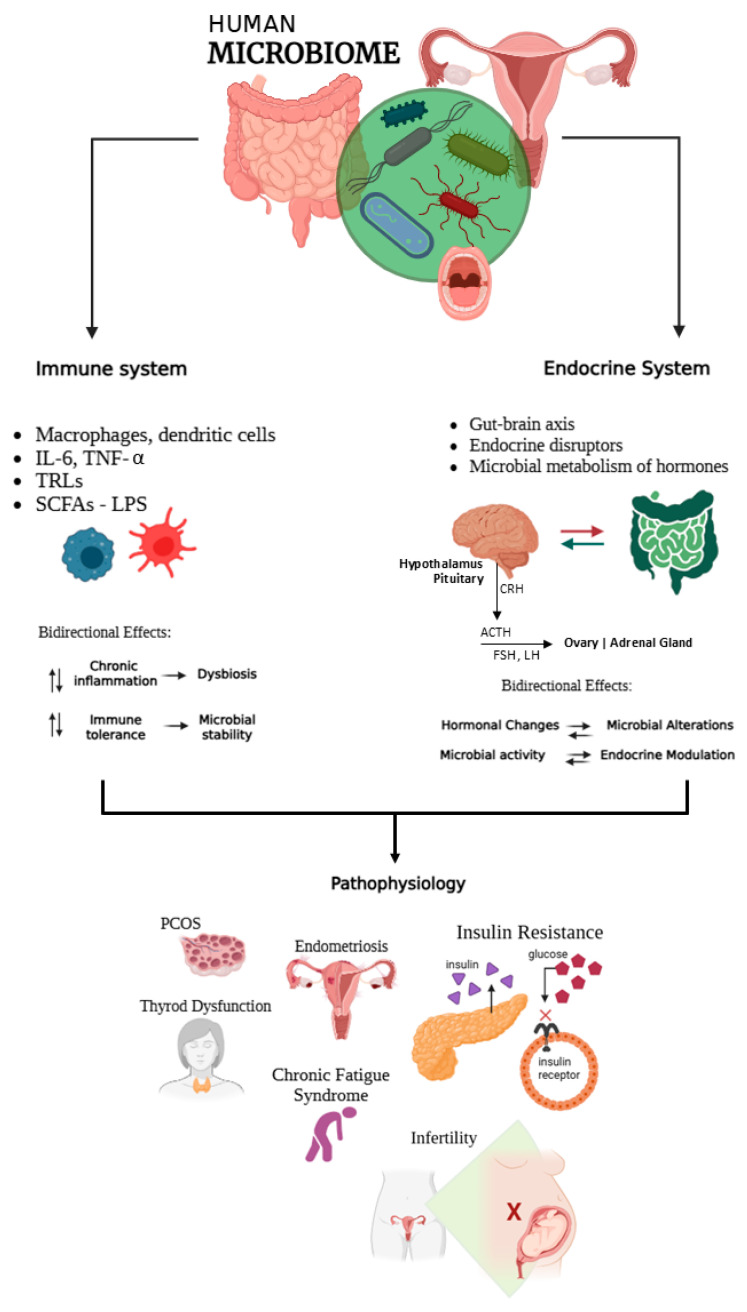
Schematic representation of the microbiome and its immuno-endocrine interactions. The diagram illustrates the bidirectional communication (represented by arrows in opposite directions) between the microbiome, the immune system (via antigen-presenting cells, cytokines, and TLR signalling), and the endocrine system. Endocrine regulation is represented through the hypothalamus and pituitary, which release corticotropin-releasing hormone (CRH) and adrenocorticotropic hormone (ACTH) to regulate the hypothalamic–pituitary–adrenal (HPA) axis, as well as gonadotropins (luteinizing hormone, LH, and follicle-stimulating hormone, FSH) that influence the hypothalamic-pituitary-gonadal (HPG) axis. These signals modulate adrenal and ovarian activity, linking microbiome-derived metabolites with hormonal control. This intricate crosstalk is involved in the pathophysiology of conditions such as PCOS, infertility, endometriosis, insulin resistance, thyroid dysfunction, and chronic fatigue syndrome. Created with BioRender by Eunice Barraza-Ortega (2025). https://www.biorender.com/ (accessed on 23 August 2025).

**Figure 2 ijms-26-08574-f002:**
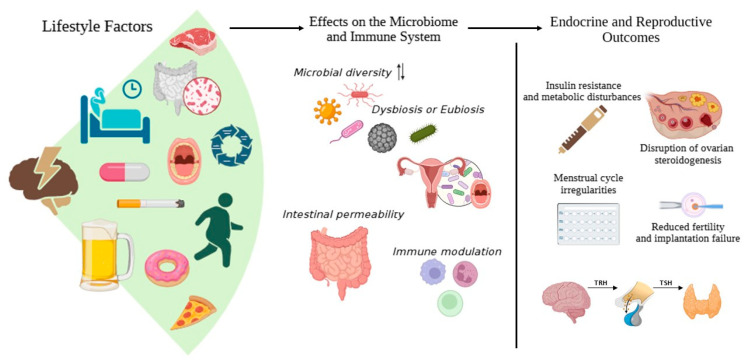
Schematic representation of how lifestyle factors affect the microbiome and endocrine system (arrows indicate directionality). Lifestyle behaviours, such as diet (including red meat consumption), physical activity, stress, and sleep, influence the gut, oral, and reproductive microbiota, thereby shaping immune responses (arrows indicate bidirectionality between microbial diversity and dysbiosis/eubiosis). These microbiome–immune interactions regulate endocrine functions through the hypothalamic-pituitary-thyroid-gonadal axis, ultimately impacting reproductive health outcomes. Created with BioRender by Eunice Barraza-Ortega (2025). https://www.biorender.com/ (accessed on 10 June 2025).

**Figure 3 ijms-26-08574-f003:**
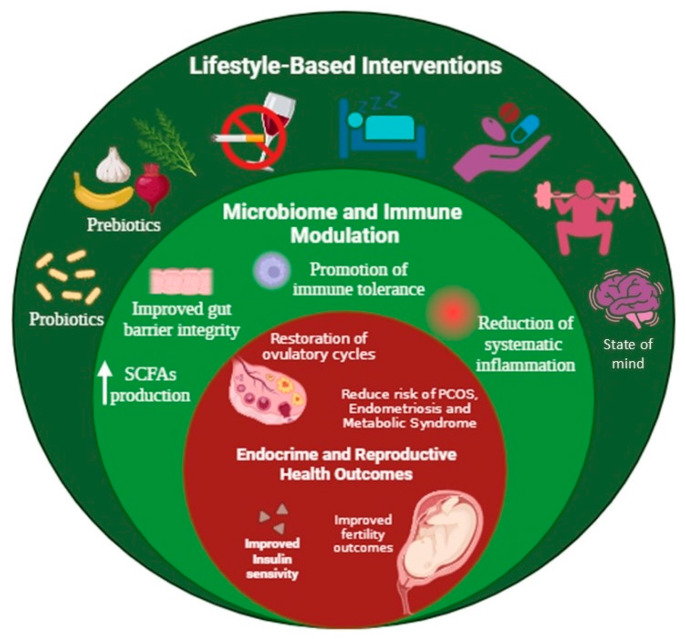
Schematic representation of preventive and therapeutic strategies based on lifestyle modifications. The diagram displays three interconnected levels: (1) lifestyle-based interventions (green), including nutrition, physical activity, stress management, sleep, and mental health; (2) the resulting modulation of the gut microbiome and immune system (olive), such as increased production of SCFAs (arrow) and reduced inflammation; and (3) subsequent improvements in endocrine and reproductive health outcomes (red), including enhanced insulin sensitivity, fertility, and a lower risk of PCOS and other related disorders. Created with BioRender by Eunice Barraza-Ortega (2025). https://www.biorender.com/ (accessed on 10 June 2025).

**Table 1 ijms-26-08574-t001:** The microbiome–-immune–-endocrine axis. Key microbial species involved in microbiome–immune–endocrine interactions and their reproductive implications. This table summarises specific microorganisms, their associated hormonal pathways, affected physiological routes, and their potential impact on reproductive health, including conditions such as PCOS, endometriosis, preterm birth, and infertility.

Microorganisms	Affected Hormones	Effects	Reproductive Health	References
*Akkermansia* *muciniphila*	Insulin/Leptin	Energetic metabolism and inflammation	Improves insulin sensitivity and reduces inflammation; important in PCOS and metabolic infertility	[90]
*Prevotella* spp.	Oestrogens and cytokines	Vaginal immunoregulation	Associated with vaginal dysbiosis and premature births if overexpressed	[59]
*Faecalibacterium prausnitzii*	Cortisol/IL-10	Systemic inflammation, gut–brain axis	Anti-inflammatory; improves mood and HPA axis. Indirectly promotes fertility by reducing stress	[91]
*Streptococcus* *agalactiae*	Does not directly affect hormones	Vaginal colonisation, neonatal infection	Increased risk of perinatal infection, chorioamnionitis and premature birth if not controlled	[92]
*Gardnerella vaginalis*	Oestrogen	Cervical mucus, vaginal environment	Associated with bacterial vaginosis; may alter embryo implantation and increase risk of miscarriage	[93]
*Clostridium* spp.	Oestrogens (oestradiol)	Oestrogen enterohepatic circulation	Increases circulating oestrogen levels; risk of hypoestrogenism, menstrual disorders, endometriosis	[94]
*Lactobacillus* spp.	Oestrogen and progesterone	Modulation of the HPG axis	Prevention of vaginal infections and premature birth; improved embryo implantation	[58]
*Bifidobacterium longum*	Cortisol	Modulation of the HPA axis	Stress reduction; promotes ovulation and success in fertility treatments	[95]
*Bacteroides* spp.	Testosterone	Hepatic metabolism of androgens	It can influence PCOS or hypogonadism	[96]
*Candida albicans*	Oestradiol modulation; inflammatory cytokines	Vaginal immunity; intestinal permeability	Associated with vaginal dysbiosis, infertility, and pregnancy complications	[97,98]
*Malassezia* spp.	Inflammatory cytokines (IL-1β, TNF-α)	Gut–immune interaction	Emerging role in mucosal inflammation and possible endocrine-metabolic dysregulation	[48]
*Human* *papillomavirus*	Immune evasion and of hormonal signalling	Cervical epithelial changes	Linked to cervical dysplasia, persistent inflammation, infertility, risk of miscarriage	[99]
*Anelloviruses*	Indirect influence via immune modulation	Vaginal virome stability; immune tolerance	Associated with *Lactobacillus*- depleted microbiota and local inflammation in CST-IV profiles; potential marker of dysbiosis	[43]
*Bacteriophages*	Indirect via hormone-sensitive bacteria	Vaginal microbial homeostasis	Dysregulation may reduce *Lactobacillus*, impair mucosal immunity, increase BV, infertility and risk of preterm birth	[100,101]
*Lactobacillus rhamnosus*	Oestrogens	Vaginal and endometrial microbiota	Promotes a *Lactobacillus*-dominant microbiome; enhances endometrial immune modulation and supports implantation	[102,103]
*Lactobacillus reuteri*	Oestrogens	Vaginal health and immune signalling	Contributes to urogenital health by reducing Group B *Streptococcus* colonisation and modulating vaginal cytokine profiles during pregnancy	[103,104]
*Chlamydia* *trachomatis*	Inflammatory cytokines (IL-6, TNF-α)	Fallopian tube mucosa	Promotes chronic inflammation leading to tubal factor infertility and ectopic pregnancy	[105]
*Toxoplasma gondii*	Th1/Th2 immune shift	Placental and foetal tissues	Acute infection during gestation disrupts placental Th1/Th2 balance → miscarriage, intrauterine growth restriction, preterm birth	[106,107]
*Mycoplasma genitalium*	Mucosal immunity	Endometrium, fallopian tubes	Linked to pelvic inflammatory disease, endometritis, infertility, tubal damage, infertility, ectopic pregnancy; ↑ risk preterm birth	[108,109]
*Ureaplasma urealyticum*	Pro-inflammatory cytokines	Amniotic cavity and cervicovaginal area	Implicated in chorioamnionitis, preterm prelabour rupture of membranes recurrent miscarriage, neonatal morbidity, and increased risk of preterm labour	[110]

## Data Availability

Not applicable.

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
