# Peer review of "The Impact of Lifestyle on Reproductive Health: Microbial Complexity, Hormonal Dysfunction, and Pregnancy Outcomes"

_ijms, 2025, doi:10.3390/ijms26178574_

Round 1

Reviewer 1 Report

Comments and Suggestions for Authors

In the present work, Eunice et al. try to review the interplay between hormonal dysfunction, the human microbiome and lifestyle factors, focusing on their impact on pregnant women and those wishing to conceive. This article aims to provide a comprehensive overview of the pathophysiological mechanisms involved and to explore preventive and therapeutic strategies. However, there are some questions that should be explained.

Major concerns

  1. In this manuscript, hypothalamic-pituitary-adrenal axis and hypothalamic-pituitary-gonadal axis are involved in hormonal regulation and dysfunctions related to reproductive health. However, hypothalamic and pituitary pay no attention, including in the figures.
  2. The ‘gut-germline axis’ has been proposed in male mice, and the author may propose it in human pregnancy.

Argaw-Denboba A, et al. Paternal microbiome perturbations impact offspring fitness. Nature. 2024;629(8012):652-659.

  1. Figure 2. In lifestyle factors, red meat should be included. In microbiome, oral cavity and reproductive organ should be added. In outcomes, hypothalamus, pituitary gland, and thyroid should be included.
  2. Figure 3. Please revise the color matching and add arrows. In addition, state of mind may have effect on pregnant women, which may be added in total manuscript.
  3. English grammar and writing style should be checked and revised throughout the manuscript.

Minor concerns

  1. Line 22, delete ‘(PCOS)’.
  2. Line 23, delete ‘(GDM)’.
  3. Line 73, delete ‘(HDACs)’.
  4. The aim of this paper should be added in the end of introduction section.
  5. Line 109, delete ‘: an overview’.
  6. Lines 169 and 407, there are two ‘human papillomavirus (HPV)’.
  7. Table 1, delete ‘(IUGR)’, ‘(PID)’, and ‘(PPROM)’.
  8. Line 350, change ‘Faecalibacterium prausnitzii’ to ‘Faecalibacterium prausnitzii (F. prausnitzii)’. Please check these throughout the manuscript, including others.
  9. For the name of bacterium, italic is enough without bold. Please check these throughout the manuscript.
  10. Line 573, delete ‘(CNS)’ and ‘(ENS)’.
Comments on the Quality of English Language

The English could be improved to more clearly express the research.

Author Response

We are thankful to all the reviewers for their constructive comments on this manuscript. We are submitting point-by-point responses to each comment. All changes in the manuscript are highlighted in yellow.

Q: In this manuscript, hypothalamic-pituitary-adrenal axis and hypothalamic-pituitary-gonadal axis are involved in hormonal regulation and dysfunctions related to reproductive health. However, hypothalamic and pituitary pay no attention, including in the figures.

R: We appreciate this valuable observation. In the revised manuscript, we explicitly describe both the HPG and HPA axes, highlighting the roles of the hypothalamus and pituitary in hormonal regulation and dysfunctions related to reproductive health. Specifically, we define the HPG axis in the Introduction and the HPA axis in Section 4, detailing the hypothalamic release of GnRH and CRH, pituitary secretion of LH/FSH and ACTH, and the downstream effects on the ovaries and adrenals. To ensure clarity, the hypothalamus and pituitary have also been added to the figures — most prominently in Figure 1, but also in Figure 2 — ensuring consistency between the text and visual representations.

Q: The ‘gut-germline axis’ has been proposed in male mice, and the author may propose it in human pregnancy.

R: We agree that the gut–germline axis is an emerging concept. We have now included a brief perspective paragraph linking paternal microbiome perturbations and germline signalling to offspring fitness, while carefully noting that human evidence during pregnancy remains preliminary. We cited a Nature article by Argaw-Denboba et al. (2024).

Q: Figure 2. In lifestyle factors, red meat should be included. In microbiome, oral cavity and reproductive organ should be added. In outcomes, hypothalamus, pituitary gland, and thyroid should be included.

R: Thank you for the suggestions. We have amended Figure 2 accordingly: (i) under Lifestyle, we added red meat; (ii) under Microbiome, we included the oral cavity and female reproductive tract compartments; and (iii) under Outcomes, we now explicitly mention the hypothalamus, pituitary gland, and thyroid. We also revised the legend to reflect these additions and clarified endocrine associations.

Q: Figure 3. Please revise the colour matching and add arrows. In addition, state of mind may have effect on pregnant women, which may be added in total manuscript.

R: We appreciate the suggestion. In Figure 3, we revised the colour scheme for better clarity, added arrows to indicate the bidirectional relationships, and incorporated “state of mind” as an additional lifestyle factor. This aligns with evidence showing that psychological well-being can influence pregnancy outcomes. The manuscript and figure legend have been updated accordingly.

Q: English grammar and writing style should be checked and revised throughout the manuscript.

R: The manuscript was proofread for English grammar and writing style. The English style is British (UK).

Q: Line 22, delete ‘(PCOS)’.

R: Corrected as requested; “(PCOS)” was removed. 

Q: Line 23, delete ‘(GDM)’.

R: “(GDM)” was removed as requested.

Q: Line 73, delete ‘(HDACs)’.

R: “(HDACs)” was removed as requested.

Q: The aim of this paper should be added in the end of introduction section.

R: We agree. An explicit aim statement has been added at the end of the Introduction to guide the reader and clarify the scope of the work.

Q: Line 109, delete ‘: an overview’.

R: Corrected as requested; “: an overview” was removed to streamline the heading.

Q: Lines 169 and 407, there are two ‘human papillomavirus (HPV)’.

R: Corrected. We retained the first full expansion in line 169 and replaced the later occurrence in line 407 with “HPV” only.

Q: Table 1, delete ‘(IUGR)’, ‘(PID)’, and ‘(PPROM)’.

R: The parenthetical acronyms in Table 1 were removed to reduce clutter.

Q: Line 350, change ‘Faecalibacterium prausnitzii’ to ‘Faecalibacterium prausnitzii (F. prausnitzii)’. Please check these throughout the manuscript, including others.

R: The sentence in line 350 was rewritten in the revised version, so this specific instance no longer appears. Nevertheless, we standardised nomenclature throughout: at first mention, we provide the full binomial with the abbreviation (e.g., Faecalibacterium prausnitzii (F. prausnitzii)), and we use the abbreviated genus thereafter. All genus/species names are italicised (no bold), consistent throughout the manuscript, tables, and figure legends.

Q: For the name of bacterium, italic is enough without bold. Please check these throughout the manuscript.

R: We removed all bold formatting from bacterial names throughout the manuscript (main text, tables, and figure legends) and ensured consistent use of italics for genus and species (e.g., Lactobacillus rhamnosus, Gardnerella vaginalis).

Q: Line 573, delete ‘(CNS)’ and ‘(ENS)’.

R: Corrected as requested; “(CNS)” and “(ENS)” were removed. The terms are written in full where relevant, and acronym use is now consistent throughout.

Reviewer 2 Report

Comments and Suggestions for Authors

The literature review provides very interesting and relevant information about the relationship between human physiology and its microbiome. The work is interesting and certainly deserves publication.

I only have a few comments:

Lines 176, 182 – “The mycome” should be replaced with “The mycobiome”

Lines 178, 181 etc. – Malassezia, Debaryomyces, Candida – it is necessary to add “spp.”

Author Response

We are thankful to all the reviewers for their constructive comments on this manuscript. We are submitting point-by-point responses to each comment. All changes in the manuscript are highlighted in yellow.

Q: Lines 176, 182 – “The mycome” should be replaced with “The mycobiome”.

R: We corrected “The mycome” to “The mycobiome” and reviewed the manuscript to ensure consistent use of “mycobiome” elsewhere.

Q: Lines 178, 181 etc. – Malassezia, Debaryomyces, Candida – it is necessary to add “spp.”

R: Thank you for the observation. All genus-level references now read Malassezia spp., Debaryomyces spp., Candida spp. throughout the manuscript (main text, tables, and figure legends).

Reviewer 3 Report

Comments and Suggestions for Authors

The authors write about how endocrine dysfunctions, such as PCOS, hypothyroidism, endometriosis, and gestational diabetes, can disrupt fertility and pregnancy outcomes. They provide a clear explanation of how lifestyle factors—nutrition, exercise, stress, sleep, and daily habits—affect both hormonal regulation and the microbiome, which together influence overall health and reproductive function. This work also highlights the rising prevalence of hormonal dysfunctions in modern societies and reviews the mechanisms, preventive approaches, and treatment strategies relevant to women who are pregnant or seeking to conceive. Overall, the manuscript is a strong contribution: it is well-organized, scientifically informative, and also accessible to a general audience.

That said, there are a few areas that could be improved:

  1. Terms such as FSH are not explained at first mention and should be defined for clarity.
  2. Table 1 and the subsequent narrative contain overlapping information; condensing these sections could reduce redundancy and improve readability.
  3. The authors should ensure that all statements are properly supported by references, and that citations are placed at the point where the information is first introduced.
  4. While the manuscript appropriately emphasizes women’s health, a brief discussion of male contributions to fertility and pregnancy outcomes would provide balance, as reproductive success is not solely determined by women.

In summary, this is a valuable manuscript that provides comprehensive insights into endocrine dysfunction and fertility, and with these refinements, it would be even stronger.

Author Response

We are thankful to all the reviewers for their constructive comments on this manuscript. We are submitting point-by-point responses to each comment. All changes in the manuscript are highlighted in yellow.

Q: Terms such as FSH are not explained at first mention and should be defined for clarity.

R: We thank the reviewer for this suggestion. In the revised manuscript, we have clarified the definitions of luteinising hormone (LH) and follicle-stimulating hormone (FSH) at their first mention within the description of the hypothalamic–pituitary–gonadal (HPG) axis. This addition improves clarity and ensures that key abbreviations are defined upon introduction.

Q: Table 1 and the subsequent narrative contain overlapping information; condensing these sections could reduce redundancy and improve readability.

R: Thank you for this helpful suggestion. In the revised version, we significantly reduced duplication by (i) streamlining Table 1 to a single, concise line per microorganism, focusing only on its primary pathway and key reproductive implications (genus/species in italics; no clinical acronyms within the table), and (ii) restructuring the subsequent text by mechanism rather than organism. Specifically, the narrative is now organised into short, non-overlapping modules (metabolic–endocrine regulators; oestrogen/androgen metabolism; vaginal ecosystem stability; pathobionts and obstetric risk; mycobiome/virome; upper-tract pathogens). Within each module, we cite one or two exemplar taxa from Table 1 without reiterating the table’s content. We also moved species-level details, which were redundant in prose, into brief parenthetical notes or citations only, and placed references at the first mention of each concept. These adjustments shorten the section and enhance readability while maintaining the essential mechanisms and clinical relevance.

Q: The authors should ensure that all statements are properly supported by references, and that citations are placed at the point where the information is first introduced.

R: We appreciate this remark. The manuscript has been carefully revised to ensure that appropriate references properly support all statements, and that citations are consistently placed at the point where the information is first introduced.

Q: While the manuscript appropriately emphasizes women’s health, a brief discussion of male contributions to fertility and pregnancy outcomes would provide balance, as reproductive success is not solely determined by women.

R: We agree. We added a concise subsection titled “Male contributions to fertility and pregnancy outcomes.” It summarises paternal metabolic/endocrine health (e.g., obesity, insulin resistance, thyroid dysfunction) and its links to semen quality, sperm DNA/epigenetic integrity, and adverse outcomes; the roles of the seminal and gut microbiomes (including the emerging gut–germline axis); and modifiable preconception factors (diet, physical activity, sleep, stress; avoidance of smoking, excess alcohol, and anabolic-androgenic steroids). This addition provides balance while keeping the review focused.

Round 2

Reviewer 1 Report

Comments and Suggestions for Authors

Thanks for author’s responses. However, there are some questions that should be explained.

  1. Line 338, correct ‘β- glucuronidase’ to ‘β-glucuronidase’.
  2. Figure 1, ‘TNF-a’; or Table 1, ‘TNF-α’.
  3. Reference 186, change ‘PLoS ONE’ to ‘PLoS One’.
Comments on the Quality of English Language

The English could be improved to more clearly express the research.

Author Response

We thank the reviewer for the detailed review. We are providing point-by-point responses.

Q: Line 338, correct ‘β- glucuronidase’ to ‘β-glucuronidase’.

R: ‘β- glucuronidase’ was changed to ‘β-glucuronidase’ as requested.

Q: Figure 1, ‘TNF-a’; or Table 1, ‘TNF-α’.

R: ‘TNF-a’ was changed to ‘TNF-α’ in Figure 1.

Q: Reference 186, change ‘PLoS ONE’ to ‘PLoS One’.

R: Reference 186 was corrected as requested.

We used Grammarly to revise English grammar and style.